# A Single-Institute Experience with C-ros Oncogene 1 Translocation in Non-Small Cell Lung Cancers in Taiwan

**DOI:** 10.3390/ijms23105789

**Published:** 2022-05-21

**Authors:** Hsiang-Sheng Wang, Chien-Ying Liu, Sheng-Chi Hsu, Shih-Chiang Huang, Tsai-Hsien Hung, Kwai-Fong Ng, Tse-Ching Chen

**Affiliations:** 1Department of Pathology, Chang Gung Memorial Hospital, Chang Gung University School of Medicine, Kwei-Shan, Taoyuan 33305, Taiwan; whs2009@cgmh.org.tw (H.-S.W.); senchi@cgmh.org.tw (S.-C.H.); ab86401112@cgmh.org.tw (S.-C.H.); nkf362@cgmh.org.tw (K.-F.N.); 2Department & Centers of Lung Cancer and Interventional Bronchoscopy, Chang Gung Memorial Hospital, Chang Gung University School of Medicine, Kwei-Shan, Taoyuan 33305, Taiwan; cyliu01@cgmh.org.tw; 3Institute of Stem Cell & Translational Cancer Research, Chang Gung Memorial Hospital at Linkou and Chang Gung University, Taoyuan 33305, Taiwan; hth0204@gmail.com

**Keywords:** *ROS1*, non-small cell lung cancer, fluorescent in situ hybridization, molecular diagnosis, crizotinib

## Abstract

(1) Background: The C-ros oncogene 1 (*ROS1*) gene translocation is an important biomarker for selecting patients for crizotinib-targeted therapy. The aim of this study was to understand the incidence, diagnostic algorithm, clinical course and objective response to crizotinib in *ROS1* translocated lung non-small cell lung cancers (NSCLCs) in Taiwan. (2) Methods: First, we retrospectively studied the *ROS1* status in 100 NSCLC samples using break-apart fluorescent in situ hybridization (FISH) and immunohistochemical (IHC) staining to establish a diagnostic algorithm. Then, we performed routine ROS1 IHC tests in 479 NSCLCs, as crizotinib was available from 2018 in Taiwan. We analyzed the objective response rate and the survival impact of crizotinib. (3) Results: Four *ROS1* translocations were clustered in epidermal growth factor receptor (*EGFR)* wild-type adenocarcinomas but not in cases with *EGFR* mutations. Strong ROS1 expression was positively correlated with *ROS1* translocation (*p* < 0.001). NSCLCs with *ROS1* translocation had a poor prognosis compared to those without *ROS1* translocation (*p* = 0.004) in the pre-crizotinib stage. Twenty NSCLCs were detected with *ROS1* translocation in 479 wild-type *EGFR* specimens from 2018. Therefore, the incidence of *ROS1* translocation is approximately 4.18% in *EGFR* wild-type NSCLCs. In these 20 *ROS1* translocation cases, 19 patients received crizotinib treatment, with an objective response rate (ORR) of 78.95% (confidence interval = 69.34% to 88.56%), including 1 complete response, 14 partial responses, 3 stable cases and 1 progressive case. Overall survival and progression-free survival were better in the 19 ROS1-translocated NSCLCs of the prospective group with crizotinib treatment than the four ROS1-translocated NSCLCs of the retrospective group without crizotinib treatment. (4) Conclusions: *ROS1*-translocated NSCLCs had a poor prognosis and could have a beneficial outcome with crizotinib.

## 1. Introduction

With the success of tyrosine kinase inhibitors (TKIs) in clinical practice, non-small cell lung cancer (NSCLC) has led to widespread research on molecular targets and testing methods [1,2,3,4]. Abnormal oncogene activation by fusion of anaplastic lymphoma kinase (*ALK*) [5,6], C-ros oncogene 1 (*ROS1*) [7,8], and Ret proto-oncogene (*RET*) [9,10] has been identified in NSCLC [11]. Crizotinib, an efficient *ALK* TKI, is effective for *ALK*-, *ROS1*- and *RET*-translocated positive patients [12,13,14]. ROS1 is a tyrosine-protein kinase that is encoded by the *ROS1* gene and has growth or differentiation functions [15]. *ROS1* translocation was first identified in lung adenocarcinoma [16], and later, the consequent *ROS1* fusion proteins were found to play significant roles in a variety of cancers, such as *FIG-ROS1* in cholangiocarcinoma [17] and inflammatory myofibroblastic tumors [16]. In addition, many partners, such as solute carrier family 34 member 2 (*SLC34A2*), cluster of differentiation 74 (*CD74*), fused in glioblastoma (*FIG*), Syndecan 4 (*SDC4*) and Ezrin (*EZR*), are rearranged with *ROS1* in NSCLC [8,18,19,20]. All these genetic arrangements consist of the 3′ ROS1 kinase domain fused to the 5′ portion of the respective partner gene and result in aberrant activation of downstream signaling and oncogenic effects [18,21]. Although the frequencies of these fusion genes are low, clinical evidence indicates that this subset of NSCLC is potentially responsive to crizotinib. Therefore, the identification of these *ROS1* translocations is critical for the optimization of personalized treatment.

To date, several molecular methods have been designed to detect *ROS1* translocation, including reverse transcription-polymerase chain reaction (RT–PCR), immunohistochemistry (IHC), and fluorescence in situ hybridization (FISH) [22]. Since lung cancers with *ROS1* translocation only account for 1–3% of NSCLCs [23,24], a detection algorithm for the *ROS1* fusion gene is important for rapid therapeutic decision. In this study, we investigated the percentage of *ROS1* translocations in NSCLC in Taiwan using IHC and FISH and attempted to establish the relationship of *ROS1* translocation with ROS1 expression and *EGFR* mutation. Moreover, we analyzed the clinicopathological features of adenocarcinomas with *ROS1* translocation, its impact on survival and the objective response rate to crizotinib.

## 2. Results

### 2.1. Retrospective Study on ROS1 Translocation in NSCLC

To determine the frequency of *ROS1* translocation in lung cancers and clinical characteristics in Taiwan, we first retrospectively examined 100 NSCLC patients treated at Lin-Kou Chang Gung Memorial Hospital, TaoYuan, Taiwan from 2010 to 2015. An equal number of male and female subjects were used in this study: 33 were current cigarette smokers, the remaining 67 had never smoked. FISH analysis with break-apart *ROS1* probes was first adopted to detect *ROS1* rearrangements (Figure 1). Among the 100 lung cancers, *ROS1* translocation was detected in four adenocarcinomas of the 100 NSCLCs (4%). Age, sex, smoking status, histology grade, and tumor stage all showed no statistically significant association with *ROS1* translocation in the study cohort (Table 1).

### 2.2. ROS1 Translocation Was Clustered in EGFR Wild-Type NSCLCs

Several studies have indicated that *ROS1* translocation usually occurs in *EGFR* wild-type adenocarcinomas. Therefore, we next studied the status of *ROS1* translocation in relation to *EGFR* mutation in these 100 cases of NSCLC. This cohort included 56 mutant *EGFR* and 44 wild-type *EGFR* tumors. The 56 mutant *EGFR* samples were composed of 26 (46.4%) exon 21 L858R mutants, 21 (37.5%) exon 19 deletions, and nine other mutations (exon 18 G719A, exon 19 L747P, exon 20 T790M, exon 20 insertion, and exon 21 L861Q, 16.1%), as shown in Table 2. All *ROS1*-translocated NSCLCs were found in the wild-type *EGFR* population with statistical significance (*p* = 0.0346).

### 2.3. ROS1 Translocation Was Associated with ROS1 Expression in IHC

Next, we tried to correlate ROS1 expression by clone D4D6 antibody with *ROS1* translocation (Figure 2). Eighty-six cases of NSCLC were scored as 0, five cases were scored as 1+, five cases were scored as 2+, and four cases were scored as 3+. All tumors with ROS1 IHC scores of 0, 1+ and 2+ had no *ROS1* translocation. In contrast, all four tumors with ROS1 IHC 3+ harbored *ROS1* rearrangements. ROS1 expression was positively associated with *ROS1* translocation (*p* < 0.0001) in lung NSCLCs (Table 3).

### 2.4. Testing Algorithm of ROS1 Tests Based on the Retrospective Study

*ROS1* translocation in NSCLCs was significantly correlated with wild-type *EGFR* and ROS1 IHC expression. For cost-effectiveness, we established a testing algorithm for NSCLCs to select eligible patients for crizotinib starting in 2018 (Figure 3). We first broke down all the test results of ROS1 IHC and FISH in Figure 3. All *ROS1* translocation cases were in ROS1 IHC 3+ NSCLCs with wild-type *EGFR*. Eighteen of 75 ROS1 IHC 3+ cases in *EGFR* mutant types had no *ROS1* rearrangement by FISH assay. Therefore, the testing algorithm was recommended as the following procedure. Each case of NSCLC was first recommended for *EGFR* mutation. If *EGFR* was wild type, then the case was tested for ROS1 and ALK by IHC. If ROS1 was IHC 3+, ROS1 FISH was performed to select eligible patients for crizotinib treatment.

### 2.5. ROS1 Translocation Was Detected Prospectively in NSCLCs

A total of 1070 NSCLCs were tested to select eligible patients for adequate targeted therapies from 2018 to 2020. They included 479 *EGFR* wild-type and 591 *EGFR* mutant specimens. Since LNCGMH switched the ROS1 antibody from clone D4D6 to clone SDP384 in 2018, we tested the clone SDP384 antibody on *EGFR* mutant cases to confirm the usability of our algorithm.

In the *EGFR* wild-type cases, 45 cases were ROS1 IHC 3+. Among the ROS1 IHC 3+ cases, 20 cases showed *ROS1* translocation with *ROS1* break-apart probes by FISH. The incidence of *ROS1* translocation was 4.18% in *EGFR* wild-type lung NSCLCs in this prospective group, and there were 75 ROS1 IHC 3+ cases in the *EGFR* mutant group. Eighteen out of these 75 cases were tested for additional *ROS1* translocation. None of them had *ROS1* translocation by the FISH assay (Figure 3).

The clinical characteristics of the 20 ROS1 FISH+/IHC 3+, FISH-/IHC 3+ 25 cases, and ROS1 IHC (-) 434 cases are shown in Table 4. The ROS1 FISH+/IHC 3+ cases had significantly higher tumor grading and later stage than ROS1 IHC (-), including 0, 1+ and 2+ cases (all *p* < 0.0001). In addition, ROS1 FISH+/IHC 3+ cases also showed a higher female prevalence than ROS1 IHC (-) cases. There were no significant differences in age, sex, stage or tumor grading when compared between FISH +/IHC 3+, FISH -/IHC 3+ and IHC (-) cases.

### 2.6. ROS1 Translocation Correlated with Responsiveness to Crizotinib

In 2018, crizotinib was available for *ROS1*-translocated NSCLCs in Taiwan. Therefore, we prospectively detected *ROS1* translocation in NSCLCs. Among the 45 ROS1 IHC 3+ patients, 19 NSCLCs with *ROS1* translocation received crizotinib treatment. One *ROS1* rearrangement patient received systemic chemotherapy (cisplatin and paclitaxel). Five of the 25 *ROS1* nontranslocated adenocarcinomas also received crizotinib treatment combined with standard chemotherapy. Two of the 25 patients received TKIs (afatinib, gefitinib and osimertinib), and 18 were treated with the standard protocol of chemotherapy only. The treatment responsiveness of crizotinib in *ROS1* translocation cases was determined by volume change at 3 months after treatment, as shown in Figure 4. The objective response rate (ORR) is shown in Table 5. The ORR to crizotinib treatment in the *ROS1* translocation group was 79% (15/19), with one complete regression. Three cases were interpreted as stable diseases, and one case progressed. The ORR of the *ROS1* nontranslocated group was 40% (2/5), and all had only partial regression. There was no statistical significance in ORR between these two groups (*p* = 0.1870). The progression-free interval was also not significantly different between these two groups. However, we compared the crizotinib-treated *ROS1* translocation group with the chemotherapy-treated *ROS1* nontranslocation group. The crizotinib-treated *ROS1* translocation group showed a better median progression-free interval (9.22 vs. 5.47) with *p* = 0.0182.

### 2.7. Prognosis of the ROS1 Translocated NSCLCs

In the pre-crizotinib stage (retrospective group), the overall 1-, 3-, 5- and 10-year rates of the 100 NSCLCs were 91%, 77%, 69% and 36% (Appendix A). Following the 8th edition of the American Joint Committee on Cancer (AJCC 8) lung cancer staging system, we defined TNM stages 1 and 2 as the early-stage group, and patients in TNM stages 3 and 4 were classified into the advanced stage group. The univariate and multivariate analyses of 10-year overall survival are shown in Table 6. There was no survival benefit across age, sex or smoking history. *ROS1* translocation (hazard ratio (HR) = 11, *p* = 0.004), higher tumor grade (HR = 3.4, *p* = 0.003) and advanced clinical stage (HR = 3.3, *p* = 0.028) were poor prognostic factors in the univariate analysis. The 10-year survival curves of different tumor grades and clinical stages are shown in Figure 5A,B. Both higher tumor grade (*p* = 0.0011) and late stage (*p* = 0.0205) showed poor survival outcomes. In multivariate analysis, *ROS1* translocation was also an independent negative factor (HR = 12.9, *p* = 0.004). Another negative indicator in multivariate analysis was higher tumor grade (HR = 3, *p* = 0.008). As shown in Figure 6A, the 10-year overall survival rate of the four *ROS1*-translocated adenocarcinoma patients was worse than that of patients without *ROS1* translocation in the pre-crizotinib period (*p* < 0.0001). In the pre-crizotinib period, two of the four *ROS1* rearranged lung cancers received standard chemotherapy and died 6 months after diagnosis. The other two cases were treated with TKI as a first-line treatment (afatinib and gefitinib). The responses were poor after three months. One of them received an additional second-line TKI treatment (osimertinib). However, the disease progressed, and the patient died within one year. The other patient was later enrolled in a crizotinib clinical trial but died at 14 months after diagnosis.

There was no significant difference in the 1-year overall survival (*p* = 0.4751) and 1-year progression-free survival (*p* = 0.6758) between *ROS1* translocated cases receiving crizotinib and *ROS1* nontranslocated cases treated with standard chemotherapy. However, as shown in Figure 6B,C, the poor prognosis related to *ROS1* translocation was reversed by crizotinib in the post-crizotinib period.

## 3. Discussion

The frequencies of *ROS1* translocation among lung NSCLC patients have been reported to be 1.7% in the USA [8], 0.7% in Japan [25], 3.1% in Korea [26], and 2–4% in China [27,28]. In this study, we found that approximately 1.9% of NSCLCs carried *ROS1* translocations in Taiwan. These results suggest that the frequency of *ROS1* translocation in NSCLC may not be affected by race differences. In addition, *ROS1* translocation was clustered in *EGFR* wild-type NSCLC from 7.1% in the pre-crizotinib stage to 4.2% in the post-crizotinib stage.

Age, sex, smoking status, and tumor stage all showed no statistically significant association with *ROS1* translocation in our retrospective analysis of 100 cases. However, *ROS1* translocation tends to be associated with a higher stage and higher tumor grading in both the pre-crizotinib stage and the post-crizotinib stage. Interestingly, the prevalence of *ROS1* translocation was significantly higher in the female group in our 479 *EGFR* wild-type cases in the prospective group. This sex-related trend was also found in other *ROS1* epidemiological studies in Europe [29] and the United Nations [24].

Conventional karyotypic analysis was regarded as unsuitable for the detection of intrachromosomal deletion and inversion events that resulted in gene translocation. Therefore, IHC- [30], RT–PCR- [19] and FISH-based [19] analyses have been used to detect *ROS1* translocation or *ROS1* fusion genes. Although IHC is the most user-friendly method for pathologists, some primary adenocarcinoma specimens were reported to have ROS1 expression without expressing the *ROS1* translocation [20,31,32,33]. Therefore, break-apart FISH should be the gold standard technique, which was also approved by the Food and Drug Agency (FDA) of the United States for the detection of *ROS1* rearrangement in NSCLC patients [34]. FISH was the technique used in the clinical trial of crizotinib treatment for NSCLC.

The diagnostic algorithm to detect *ROS1* translocation remains under debate. A previous study suggested that all cases with wild-type *EGFR* should be tested for *ROS1* translocation with a FISH assay [26]. In our study, *ROS1* translocation was also mutually exclusive to *EGFR* mutations in both the pre-crizotinib stage (clone D4D6) and the post-crizotinib stage (clone SDP384). This finding was the same as that in previous reports [35]. In the previous correlation of ROS1 IHC with FISH, clone D4D6 had 88% sensitivity and 98% specificity [36], while clone SDP384 showed 98% sensitivity and 76% specificity [37]. The lower specificity rate of clone SDP384 was compatible with our prospective cases, in which 25 out of 45 ROS1 IHC 3+ cases had no *ROS1* translocation in FISH assay.

Overall, only a small proportion of *EGFR* wild-type lung cancers harbored *ROS1* translocation. Even in the ROS1 IHC 3+ groups, only 20 of 45 cases had *ROS1* rearrangements in our study. None of the ROS1 IHC 0, IHC 1+, or IHC 2+ cases were found to have *ROS1* translocation. For cost-effectiveness, we proposed the testing algorithm shown in Figure 3, in which only ROS1 IHC 3+ cases will be performed for FISH study in the *EGFR* wild-type group. It was also found that *ROS1* translocation was an independent poor prognostic factor in several studies [28,38]. We also had the same finding in our retrospective groups.

In the pre-crizotinib stage, some NSCLCs with *ROS1* translocation might be treated with TKIs, even without *EGFR* mutations. Most TKIs, such as erlotinib or gefitinib, target *EGFR* mutations. Although the *ROS1* gene also belongs to the receptor tyrosine kinase family, its molecular structure is different from that of *EGFR*. Therefore, erlotinib and other TKIs showed little or no treatment effect on tumors harboring *ROS1* mutations in previous studies [39,40]. In our retrospective group, two *ROS1*-rearranged lung cancers were also refractory to first- and second-line TKIs (afatinib, gefitinib and osimertinib).

In 2018, crizotinib was included on the National Health Insurances of Taiwan payment list for the treatment of *ALK*- and *ROS1*-translocated NSCLCs. Therefore, 19 of the *ROS1*-translocated cases received standard crizotinib treatment. The objective response rate was 79%, which was slightly higher than the rates in the European [12,41,42] (72–76%), Pacific [43] (72%) and Asian regions [44] (72%), and similar to or lower than the rates in China (74–94%). In our study, the complete remission rate was 5%, which was lower than those in Europe [12] (17%) and Japan [43] (11%) but higher than that in China [18] (0%). The partial regression rate was 74%, which was slightly higher than other studies (58–72%). The percentages of stable disease ranged from 14% to 21% within 6 months in most of the studies. The rate of stable disease was 16% in our study. The rate of progressive disease ranged from 0 to 10% and was 5% after 3 months of treatment with crizotinib in our study. Thus, the effectiveness of crizotinib is similar regardless of race and region. [12,18,41,42,43]

The progression-free and overall survival durations of *ROS1* translocation cases under crizotinib treatment were not significantly different from those of *ROS1* nontranslocation cases under chemotherapy in the present study. However, we observed a trend of better progression-free survival. In stage 4 patients, the one-year median progression-free durations of *ROS1* translocation cases under crizotinib treatment and *ROS1* nontranslocation cases under chemotherapy were 9.22 months (8.82–9.62) and 5.17 months (4.52–6.41), respectively (*p* = 0.0182). These results are similar to those of the European crizotinib clinical trial, with progression-free durations of 9.1 months for *ROS1* translocation in patients treated with crizotinib vs. 7.2 months for the chemotherapy group [12].

The responsiveness in two lung cancers without *ROS1* translocation to crizotinib also led to no statistical significance between NSCLCs with and without *ROS1* translocation. This might be due to the presence of mutations that were responsive to crizotinib, such as *RET* translocation, in responsive lung cancers. However, *ROS1* translocation cases showed a significantly poor overall survival rate in the pre-crizotinib stage. Therefore, crizotinib could reverse the poor prognosis of *ROS1*-translocated lung cancers in the crizotinib stage in Taiwan.

## 4. Materials and Methods

### 4.1. Tissue Specimens

One hundred paraffin-embedded specimens from patients with lung cancer who underwent surgery for lung cancer at Lin-Kou Chang Gung Memorial Hospital (LKCGMH) from 2010 to 2015 were enrolled in the retrospective study. Clinicopathological data were obtained from patient charts. Another 1070 lung tissue specimens from 2018 to 2020 were prospectively enrolled for this ROS1 study. The study was performed with the approval of the Institutional Review Board.

### 4.2. Fluorescence in Situ Hybridization Assay

To assess rearrangements in the *ROS1* loci, a novel break-apart FISH assay was designed. For the 3′ *ROS1* probe, the bacterial artificial chromosome clone used was RP11-1036C2, which was labeled with SpectrumGreen-dUTP (Abbott Molecular/Vysis, Des Plaines, IL, USA). For the 5′ *ROS1* probe, the bacterial artificial chromosome clones were SpectrumOrange-dUTP-labeled RP11-835I21. Tissue sections (4 μm thick) were placed onto coated slides, air-dried, and baked overnight at 56 °C. FISH analysis was performed as previously described. Slides were analyzed using a multifiltered fluorescence microscope (Olympus BX61, Southall, Middlesex, UK) following standard procedures. A minimum of 100 cells was scored. When 15% or more single nuclei were found to harbor split (break-apart) signals, the case was classified as positive. We compared the results with our in-house *ROS1* break-apart probes with Vysis *ROS1* break-apart FISH probes (Abbott Molecular/Vysis, Des Plaines, IL, USA) in 20 lung cancers. The results completely matched.

### 4.3. EGFR Mutational Analysis

The tumor purity of each sample was examined by pathologists. For cases with less than 25% purity, manual microdissection was performed to enrich the tumor purity. FFPE tissue genomic DNA from each sample was extracted using the QIAamp DNA FFPE Tissue Kit (Qiagen, Hilden, Germany) and was then used for the PCR amplification of *EGFR* exons 18, 19, 20, and 21 as previously reported. The PCR products were purified for Sanger sequencing [45].

### 4.4. Immunohistochemistry

ROS1 staining was performed on 4-micrometer paraffin-embedded sections of NSCLC specimens by using the Leica Bondmax system. When using the Leica Bondmax stainer (Leica Biosystems, Wetzlar, Germany), antigen retrieval was performed at 100 °C for 40 min in ER2 solution (ethylenediaminetetraacetic acid (EDTA), pH 9.0). Anti-ROS1 antibodies (Clone D4D6 (Cell Signaling, Danvers, MA, USA) before 2018 and clone SDP384 (Ventana Medical Systems Inc, Tucson, Arizona) after 2018) were used in IHC with the Bond Polymer Refine Detection/polymeric horseradish peroxidase (HRP)-linker antibody conjugate system (Leica Biosystems). IHC scoring was performed as previously described [46]. In brief, IHC 3+ cases typically revealed discernible strong cytoplasmic positivity. IHC 2+ cases showed readily recognizable positivity but with a lower intensity of staining than that seen in IHC 3+ cases. IHC 1+ cases represented faint positivity, with a lower intensity of staining than seen in IHC 2+ cases. Negative cases did not show any staining. ROS1 scoring was conducted according to the criteria described by the manufacturer.

### 4.5. Statistical Analysis

The correlations between *ROS1* translocation, ROS1 expression, and clinical features were assessed by the chi-square test. Survival curves were plotted by the Kaplan–Meier method, with the log-rank test applied for comparison. The Cox proportional hazards regression model was employed to evaluate the independent prognostic factors. All tests were two-sided, and *p* values < 0.05 were considered statistically significant. The statistical analyses were performed with Prism 5.0 (GraphPad Software, La Jolla, CA, USA) and SPSS (SPSS Inc., Chicago, IL, USA) software.

## 5. Conclusions

The incidence of *ROS1* rearrangement was approximately 4% in *EGFR* wild-type NSCLC in Taiwan. For cost-effectiveness, it was recommended to test *ROS1* translocation in *EGFR* wild-type NSCLCs. IHC was useful for selecting ROS1 3+ specimens for FISH assay. The ORR to crizotinib in NSCLC was 79% in Taiwan. *ROS1* translocated lung cancer is an independent poor prognostic overall survival factor and could be reversed by crizotinib treatment.

## Figures and Tables

**Figure 1 ijms-23-05789-f001:**
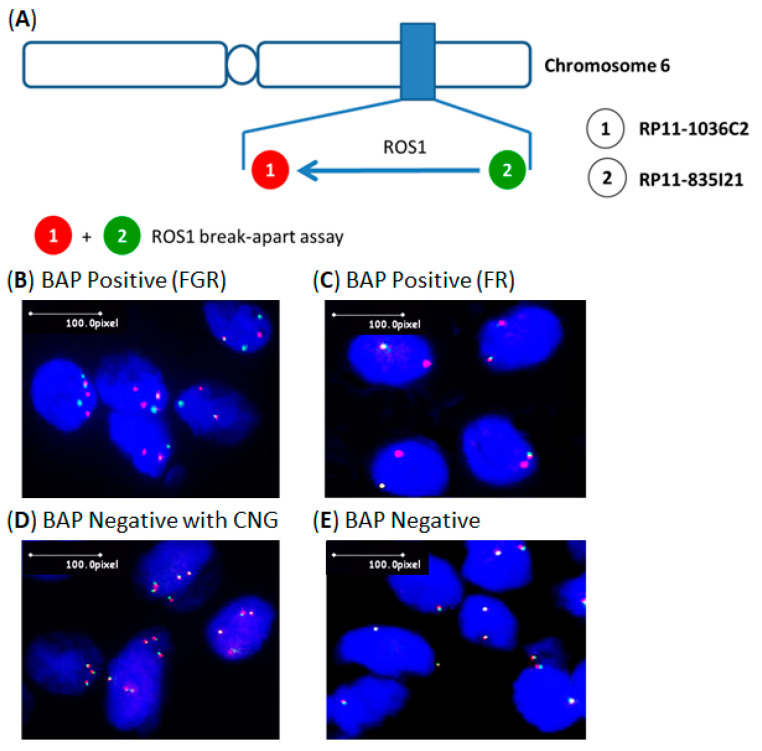
Detection of *ROS1* translocation genes by FISH analysis. (**A**) Scheme of *ROS1* break-apart probe construction. (**B**) Positive *ROS1* translocation sample with classic combination of fused, green, and red signals (FGR pattern). (**C**) Positive *ROS1* translocation sample with loss of isolated green signal (FR pattern). (**D**) Positive *ROS1* translocation sample with copy number gain (CNG). (**E**) Negative *ROS1* translocation sample.

**Figure 2 ijms-23-05789-f002:**
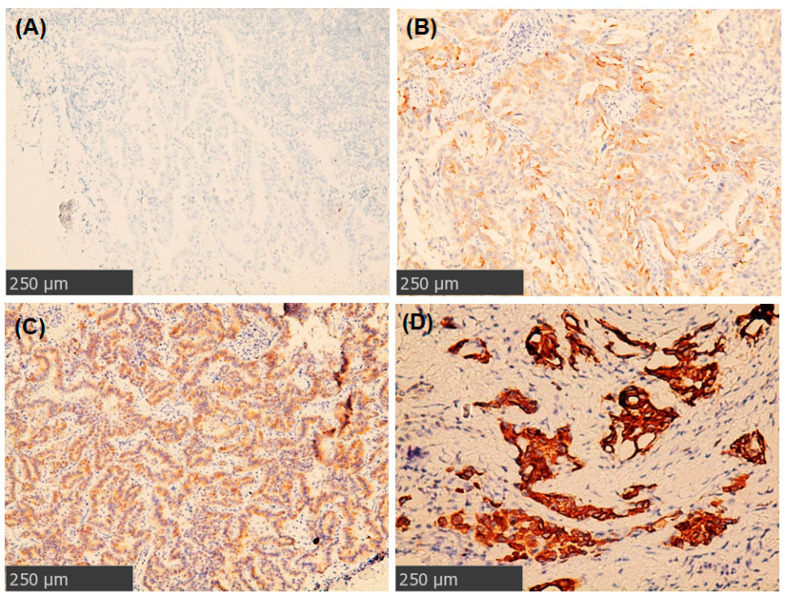
Grading of ROS1 protein overexpression by IHC staining. (**A**) A patient with a negative stain for ROS1 (0). (**B**) A case with weak cytoplasmic staining (1+) for ROS1. (**C**) A case with moderate (2+) cytoplasmic stain for ROS1. (**D**) A patient with strong (3+) granular cytoplasmic staining for ROS1.

**Figure 3 ijms-23-05789-f003:**
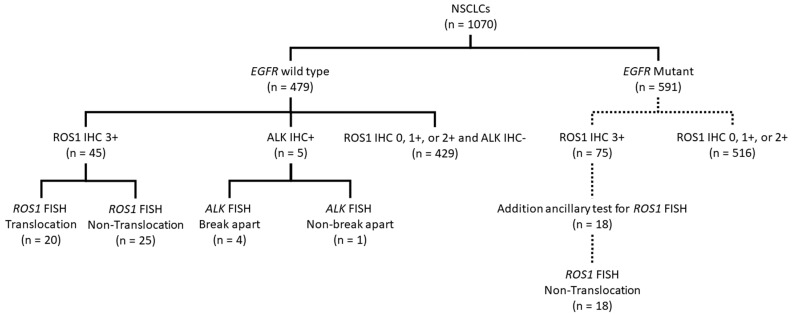
Testing algorithm for selecting *ROS1* translocation for crizotinib treatment. A total of 1070 cases were collected from 2018 to 2020 with 479 *EGFR* wild-type and 591 *EGFR* mutant. In *EGFR* wild-type cases, 45 cases showed ROS1 IHC 3+, which included 20 cases that harbored *ROS1* translocation. There were 75 cases of ROS1 IHC 3+ in the *EGFR* mutant group. Eighteen out of these 75 cases are also tested for *ROS1* translocation, and none of them had *ROS1* translocation. For cost effectiveness, each case of NSCLC was first tested for *EGFR* mutation assay. If *EGFR* was wild type, then the case would be tested for ROS1 and ALK by IHC. If the ROS1 was IHC 3+, *ROS1* FISH would be performed to select eligible patients for crizotinib treatment.

**Figure 4 ijms-23-05789-f004:**
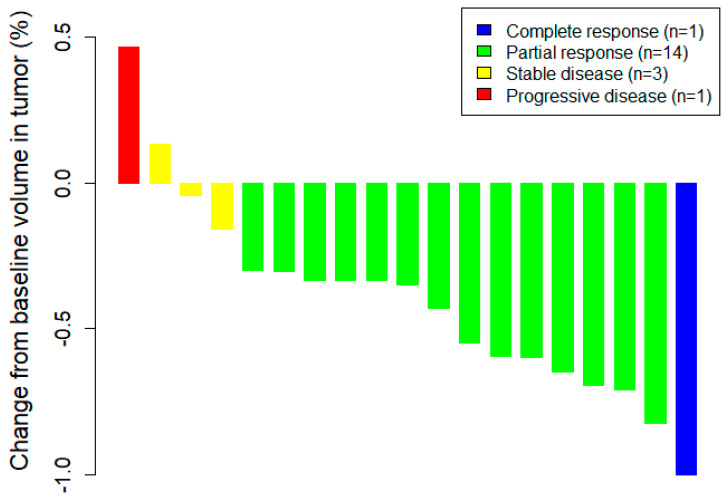
Waterfall plots of *ROS1* translocated cases treated with crizotinib. A total of 19 *ROS1* rearranged NSCLC patients received crizotinib treatment, including 1 case with complete remission, 14 cases showing partial regression, 3 cases under stable disease, and 1 case rapidly progressing.

**Figure 5 ijms-23-05789-f005:**
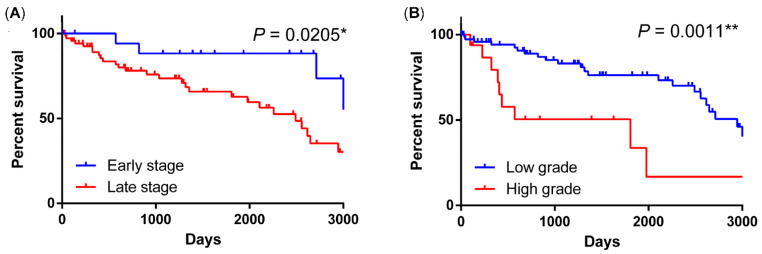
Overall survival with different stage and tumor grade in pre-crozotinib stage. (**A**) Late stage showed a significant worse outcome. (**B**) High tumor grade also indicated a poor prognostic value. * *p* < 0.05, ** *p* < 0.01.

**Figure 6 ijms-23-05789-f006:**
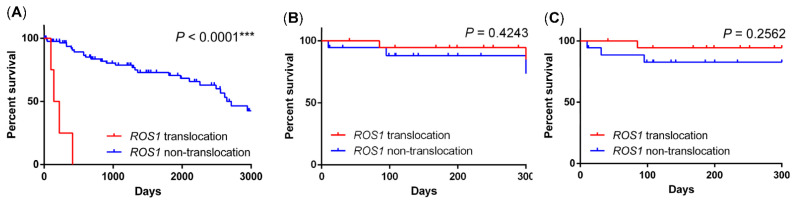
Overall survivals of non-small cell lung cancers treated with different treatment modalities. The overall survival of *ROS1* translocated lung cancers had a poor overall l survival as compared to those without *ROS1* translocation (**A**). The crizotinib treatment reverse the poor prognostic outcome from the pre-crizotinib period ((**A**), *p* < 0.0001) to no significant difference in crizotinib stage ((**B**), *p* = 0.4243). The progression-free survival in the post-crizotinib period shows no significant difference between *ROS1* translocated cases treated with crizotinib and *ROS1* non-translocated cases treated with standard chemotherapy (**C**). *** *p* < 0.001.

**Table 1 ijms-23-05789-t001:** Characteristics of *ROS1* translocated and non-translocated groups in pre-crizotinib period.

		*ROS1* Translocated, *n* = 4	*ROS1* Non-Translocated, *n* = 96	*p* Value
Gender, *n* (%)	Male	2 (50)	48 (50)	
	Female	2 (50)	48 (50)	1.000
Age, *n* (%)	≥65	0 (0)	44 (46)	
	<65	4 (100)	52 (54)	0.1283
Smoking, *n* (%)	Yes	1 (25)	32 (34)	
	No	3 (75)	64 (70)	1.000
Clinical Stage, *n* (%)	1	0 (0)	14 (15)	
	2	0 (0)	5 (5)	
	3	0 (0)	15 (16)	
	4	4 (100)	62 (65)	0.5426
Tumor grade, *n* (%)	Well differentiated	0 (0)	28 (29)	
	Moderate differentiated	3 (75)	47 (49)	
	Poorly differentiated	1 (25)	20 (21)	
	Undifferentiated	0 (0)	1 (1)	0.6235

**Table 2 ijms-23-05789-t002:** Relationship between *ROS1* and *EGFR* in lung cancers.

		*ROS1* Translocated, *n* = 4	*ROS1* Non-Translocated, *n* = 96	*p* Value
*EGFR*, *n* (%)	Wild type	4 (100)	40 (42)	
	Mutant	0 (0)	56 (58)	0.0346

**Table 3 ijms-23-05789-t003:** Association of ROS1 expression with *ROS1* translocation.

	IHC ROS1 Expression				
FISH *ROS1* Pattern	0 *n* = 86	1+ *n* = 6	2+ *n* = 4	3+ *n* = 4	*p* Value
Rearrangement, *n* (%)	0 (0)	0 (0)	0 (0)	4 (100)	
Non-rearrangement, *n* (%)	86 (100)	6 (100)	4 (100)	0 (0)	<0.0001

**Table 4 ijms-23-05789-t004:** Demographics and clinical characteristics in *E**GFR* wild-type prospective cases.

		*EGFR* Wild Type			*p* Value
		FISH(+)/IHC(3+), *n* = 20	FISH(-)/IHC(3+), *n* = 25	IHC(0,1+,2+), *n* = 434	
Gender, *n* (%)	Male	3 (15)	9 (36)	231 (53)	
	Female	17 (85)	16 (64)	203 (47)	0.0012
Age, *n* (%)	≥65	9 (45)	12 (48)	178 (41)	
	<65	11 (55)	13 (52)	256 (59)	0.7491
Smoking, *n* (%)	Yes	5 (25)	8 (32)	113 (26)	
	No	15 (75)	17 (68)	321 (74)	0.7977
Stage, *n* (%)	1	0 (0)	0 (0)	113 (26)	
	2	0 (0)	2 (8)	79 (18)	
	3	0 (0)	2 (8)	64 (15)	
	4	20 (100)	21 (84)	178 (41)	<0.0001
Tumor grade, *n* (%)	Well differentiated	0 (0)	5 (20)	149 (34)	
	Moderate differentiated	6 (30)	8 (32)	225 (52)	
	Poorly differentiated	14 (70)	11 (44)	57 (13)	
	Undifferentiated	0 (0)	1 (4)	3 (1)	<0.0001

**Table 5 ijms-23-05789-t005:** Objective response rate of different treatment in ROS1 IHC 3+ cases.

		ROS1 IHC 3+, *n* = 45
		*ROS1* FISH Translocation, *n* = 20	*ROS1* FISH Non-Translocation, *n* = 25
		Crizotinib, *n* = 19	Chemotherapy, *n* = 1	Crizotinib, *n* = 5	CHEMOTHERAPY, *n* = 18	TKI, *n* = 2
Response	CR, *n* (%)	1 (5)	0 (0)	0 (0)	0 (0)	0 (0)
PR, *n* (%)	14 (74)	0 (0)	2 (40)	5 (28)	1 (50)
SD, *n* (%)	3 (16)	1 (100)	1 (20)	9 (50)	1 (50)
PD, *n* (%)	1 (5)	0 (0)	2 (40)	4 (22)	0 (0)
ORR, *n* (%)	15 (79)	0 (0)	2 (40)	5 (28)	1 (50)
Median progression free survival, month (95% CI)	9.22 (8.82–9.62)	2.74 (2.74–2.74)	8.06 (5.25–10.88)	5.47 (4.52–6.41)	9.53 (0–19.65)
	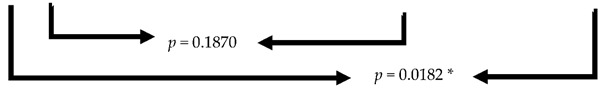	

ORR: objective response rate, TKI: receptor tyrosine kinase inhibitor, CI: confidence interval, CR: complete response, PR: partial response, SD: stable disease, PD: progressive disease. * *p* < 0.05.

**Table 6 ijms-23-05789-t006:** Univariate and multivariate analysis for hazard ration in overall survival.

	Univariate Analysis			Multivariate Analysis		
	HR	95% CI	*p* Value	HR	95% CI	*p* Value
Age (≥65)	1.1	0.54–2.2	0.817	1.2	0.54–2.5	0.699
Gender (Male)	1.3	0.65–2.6	0.457	1.1	0.36–3.4	0.865
Smoking (Yes)	1.6	0.75–3.4	0.224	1.4	0.40–4.7	0.615
*ROS1* (Translocation)	11	2.1–54	0.004	12.9	2.22–74.3	0.004
Tumor grade (High)	3.4	1.5–7.5	0.003	3	1.33–6.8	0.008
Advanced stage (stage 3 or 4)	3.3	1.1–9.3	0.028	2.9	0.98–8.5	0.055

HR: hazard ratio, CI: confidence interval.

## Data Availability

The raw data supporting the conclusions of this article will be made available by the authors, at a reasonable request.

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
