# Peer review of "A Single-Institute Experience with C-ros Oncogene 1 Translocation in Non-Small Cell Lung Cancers in Taiwan"

_ijms, 2022, doi:10.3390/ijms23105789_

Round 1

Reviewer 1 Report

  1. Abstract Page 1-41:  This statement does not need "In Taiwan".
  2. Abstract: Is the number of ROS1 translocated NSCLC without crizotinib treatment 5? (retrospective case 4; propspective case 1) It should be clarified when the comparison with 19 patients with crizotinib treatment is desribed.    
  3.  Figure 1 legend: The full term of FGR and FR should be marked.
  4. Table 4: There are 3 columns, so the p value should be explained, representing the comparison of which columns. 

Author Response

Point 1.

Abstract Page 1-41:  This statement does not need "In Taiwan".

Response 1.

We delete it from the sentence.”

Point 2.

Abstract: Is the number of ROS1 translocated NSCLC without crizotinib treatment 5? (retrospective case 4; propspective case 1) It should be clarified when the comparison with 19 patients with crizotinib treatment is described.   

Response 2.

Yes, the number of ROS1 translocated NSCLC without crizotinib treatment 5 (retrospective group 4; propspective group 1). We found out that crizotinib could reverse the poor prognostic result in the retrospective group (4) without crizotinib treatment. Therefore, we revise the text “Overall survival and progression-free survival were better in ROS1-translocated NSCLCs with targeted treatment than in ROS1-translocated NSCLCs without crizotinib treatment” into “Overall survival and progression-free survival were better in the 19 ROS1-translocated NSCLCs of the prospective group with crizotinib treatment than the 4 ROS1-translocated NSCLCs of retrospective group without crizotinib treatment.”

Point 3.

Figure 1 legend: The full term of FGR and FR should be marked.

Response 3.

We added the detail description of FGR and FR in the main text. We change description in Figure 1 “(B) Positive ROS1 translocation sample with FGR pattern. (C) Positive ROS1 translocation sample with FR pattern.” into “(B) Positive ROS1 translocation sample with classic combination of fused, green, and red signals (FGR pattern). (C) Positive ROS1 translocation sample with lost of isolated green signal (FR pattern).”

Point 4.

Table 4: There are 3 columns, so the p value should be explained, representing the comparison of which columns.

Response 4.

We revise our description for Table 4 from “The clinical characteristics of the 20 ROS1 FISH+/IHC 3+, FISH-/IHC 3+ 25 cases, and ROS1 IHC (-) 434 cases are shown in Table 4. ROS1 IHC 3+ cases had significantly higher tumor grading and later stage than ROS1 IHC (-), including 0, 1+ and 2+ cases (all P < 0.0001). In addition, IHC 3+ cases also showed a higher female prevalence than ROS1 IHC (-) cases. There were no significant differences in age, sex, stage or tumor grading between IHC 3+/FISH +, and IHC 3+/FISH - cases.” into “The clinical characteristics of the 20 ROS1 FISH+/IHC 3+, FISH-/IHC 3+ 25 cases, and ROS1 IHC (-) 434 cases are shown in Table 4. ROS1 FISH+/IHC 3+ cases had significantly higher tumor grading and later stage than ROS1 IHC (-), including 0, 1+ and 2+ cases (all P < 0.0001). In addition, ROS1 FISH+/IHC 3+ cases also showed a higher female prevalence than ROS1 IHC (-) cases. There were no significant differences in age, sex, stage or tumor grading when compared between FISH +/IHC 3+, FISH -/IHC 3+ and IHC (-) cases.”

Reviewer 2 Report

Review of ijms 1729788-v1

A single-institute experience with ROS1 translocation in non-small cell lung cancers in Taiwane

This is a fairly well written manuscript . The manuscript readability and thus comprehensibility could be improved by the judicious use of abbreviations. It is not mandatory to use abbreviations. Abbreviations are normally used to cut down on the size of manuscripts especially when terms are repeated frequently. The authors should be aware that excessive use of abbreviated terms reduces the readability of a manuscript especially if the reader is not familiar with the terminology. In such a small manuscript as the one submitted I wonder how essential all of the abbreviations are. A few could be removed to considerably improve the flow of the manuscript to improve its comprehension.

I only detected a few minor changes in the manuscript , these are listed below along with additional suggestions.

Changes required

Title Taiwane should be Taiwan, it would be better to spell out the complete ROS1 term.

The authors should provide institutional e-mail contact information rather than private e-mails.

Line 55 Please ensure that all abbreviated terms are defined at their first point of use in the text. A list of abbreviations in the manuscript may also be useful.

Line 77 “patients included of 50 males and 50 females,” delete ‘of’

Suggestion

An equal number of male and female subjects were used in this study, 33 were current cigarette smokers, the remaining 67 had never smoked.

Table 2 mutate should be mutant

Please note the accepted use of P as a statistical term is as a CAP italic term ie P

Figure 4 provide an explanation of the abbreviated terms CR, PR, SD, PD in the colour key in the legend to this figure as provided in Table 5.

Table 5 CI should be confidence interval

Line 197 define the abbreviations

Table 6   HR-hazard ratio, CI confidence interval

Figure 5 and elsewhere in manuscript    The authors need to select clearer line forms to the ones indicated in the figures. Maybe coloured lines would be better.

Line 239 2-4% is clearer

Line 283 Re-word “In early 2018, crizotinib started to be on the payment list of the National Health Insurances of Taiwan for ALK- and ROS1-translocated NSCLC treatment. Maybe In 2018, crizotinib was listed on the National Health Insurances of Taiwan payment list for the treatment for ALK- and ROS1-translocated NSCLC. would be better.

Line 384 “without undue reservation” maybe at a reasonable request would be better.

The bibliography needs to be in the form

  1. Hauser, P.;Wang, S.; Didenko, V.V. Apoptotic Bodies: Selective Detection in Extracellular Vesicles. Signal Transduct. Immunohistochem. 2017, 1554, 193–200.
  2. Kowal, J.; Arras, G.; Colombo, M.; Jouve, M.; Morath, J.P.; Primdal-Bengtson, B.; Dingli, F.; Loew, D.; Tkach, M.; Théry, C. Proteomic comparison defines novel markers to characterize heterogeneous populations of extracellular vesicle subtypes. Proc. Natl. Acad. Sci. USA 2016, 113, E968–E977. [CrossRef]
  3. Fitts, C.A.; Ji, N.; Li, Y.; Tan, C. Exploiting Exosomes in Cancer Liquid Biopsies and Drug Delivery. Adv. Healthc. Mater. 2019, 8,e1801268. [CrossRef]

ie the journal name is italicized and not the title of the study

Author Response

Point 1.

Title Taiwane should be Taiwan, it would be better to spell out the complete ROS1 term.

Response 1.

We correct the title “A single-institute experience with ROS1 translocation…” into “A single-institute experience with C-ros oncogene 1 translocation…in Taiwan”

Point 2.

The authors should provide institutional e-mail contact information rather than private e-mails.

Response 2.

We change the email from “[email protected]” into “[email protected]

Point 3.

Line 55 Please ensure that all abbreviated terms are defined at their first point of use in the text. A list of abbreviations in the manuscript may also be useful.

Response 3.

We do not use AC to stand for “adenocarcinoma”,  use BAC to represent “bacterial artificial chromosome” and add a list of abbreviations at the end of our manuscript.

Point 4.

Line 77 “patients included of 50 males and 50 females,” delete ‘of’

Suggestion

An equal number of male and female subjects were used in this study, 33 were current cigarette smokers, the remaining 67 had never smoked.

Response 4.

We revise our text from “The patients included of 50 males and 50 females, with 67 never smokers and 33 smokers.” into “An equal number of male and female subjects were used in this study, 33 were current cigarette smokers, the remaining 67 had never smoked.”

Point 5.

Table 2 mutate should be mutant

Response 5.

We revise the “mutate” into “mutant” in Table 2.

Point 6.

Please note the accepted use of P as a statistical term is as a CAP italic term ie P

Response 6.

All P value within our manuscript and Figures changed from “p” into “P

Point 7.

Figure 4 provide an explanation of the abbreviated terms CR, PR, SD, PD in the colour key in the legend to this figure as provided in Table 5.

Response 7.

The abbreviated terms CR, PR, SD and PD in Figure 4 are revised to “complete response, partial response, stable disease and progressive disease”

Point 8.

Table 5 CI should be confidence interval

Response 8.

We revise the text “confident interval” in Table 5 into “confidence interval”

Point 9.

Line 197 define the abbreviations

Response 9.

We revise the text “Following the AJCC 8 lung cancer staging system…” to “Following the 8th edition of the American Joint Committee on Cancer (AJCC 8) lung cancer staging system…”

Point 10.

Table 6   HR-hazard ratio, CI confidence interval

Response 10.

We revise the words in Table 6 from “confident interval” to “confidence interval” and from “hazard ration” to “hazard ratio”

Point 11.

Figure 5 and elsewhere in manuscript The authors need to select clearer line forms to the ones indicated in the figures. Maybe coloured lines would be better.

Response 11.

We use red and blue lines to represent the survival curves in Figure 5 and Figure 6.

Point 12.

Line 239 2-4% is clearer

Response 12.

We revise the text from “…and 2.0%-4% in China…” to “…and 2-4% in China…”

Point 13.

Line 283 Re-word “In early 2018, crizotinib started to be on the payment list of the National Health Insurances of Taiwan for ALK- and ROS1-translocated NSCLC treatment. Maybe In 2018, crizotinib was listed on the National Health Insurances of Taiwan payment list for the treatment for ALK- and ROS1-translocated NSCLC. would be better.

Response 13.

We change the text from “…In early 2018, crizotinib started to be on the payment list of the National Health Insurances of Taiwan for ALK- and ROS1-translocated NSCLC treatment.…” to “…In 2018, crizotinib was included on the National Health Insurances of Taiwan payment list for the treatment of ALK- and ROS1-translocated NSCLCs…”

Point 14.

Line 384 “without undue reservation” maybe at a reasonable request would be better.

Response 14.

We revise the text from “…without undue reservation” to “…at a reasonable request”

Point 15.

The bibliography needs to be in the form

Response 15.

We download the EndNote styles of MDPI ACS Journals from https://endnote.com/style_download/mdpi/ and update the bibliography
